# Electrochemical Synergies of Heterostructured Fe_2_O_3_-MnO Catalyst for Oxygen Evolution Reaction in Alkaline Water Splitting

**DOI:** 10.3390/nano9101486

**Published:** 2019-10-18

**Authors:** Junyeong Kim, Jun Neoung Heo, Jeong Yeon Do, Rama Krishna Chava, Misook Kang

**Affiliations:** Department of Chemistry, College of Natural Sciences, Yeungnam University, Gyeongsan, Gyeongbuk 38541, Korea; jjun@ynu.ac.kr (J.K.); hjn2521@naver.com (J.N.H.); drcrkphysics@hotmail.com (R.K.C.)

**Keywords:** heterostructured Fe_2_O_3_-MnO, oxygen evolution reaction, alkaline water splitting, electrochemical synergy

## Abstract

For efficient electrode development in an electrolysis system, Fe_2_O_3_, MnO, and heterojunction Fe_2_O_3_-MnO materials were synthesized via a simple sol–gel method. These particles were coated on a Ni-foam (NF) electrode, and the resulting material was used as an electrode to be used during an oxygen evolution reaction (OER). A 1000-cycle OER test in a KOH alkaline electrolyte indicated that the heterojunction Fe_2_O_3_-MnO/NF electrode exhibited the most stable and highest OER activity: it exhibited a low overvoltage (n) of 370 mV and a small Tafel slope of 66 mV/dec. X-ray photoelectron spectroscopy indicated that the excellent redox performance contributed to the synergy of Mn and Fe, which enhanced the OER performance of the Fe_2_O_3_-MnO/NF electrode. Furthermore, the effective redox reaction of Mn and Fe indicated that the structure maintained stability even under 1000 repeated OER cycles.

## 1. Introduction

Hydrogen has the highest energy density per unit mass and is known as a clean energy source that does not emit any exhaust gas except for water when used as a fuel. Although significant amounts of H_2_ have been produced through the modification of fossil fuels [1], global warming and depletion of fossil fuels have recently led to attempts to obtain H_2_ via new conversion methods in various countries around the world. In particular, the method of electrolyzing water to produce H_2_ is an innovative and ecofriendly technology because it uses the cleanest and abundant resource on earth—water—as a raw material for H_2_ [2]. The water electrolysis reaction, which is described by 2H_2_O ↔ H_2_ + O_2_, consists of two half reactions: the hydrogen evolution reaction (HER) and the oxygen evolution reaction (OER) [3]. HERs are one-electron reactions, and catalysts that exhibit excellent activity have been reported because of the relatively simple mechanism of the reaction: Pt electrodes with a high reduction potential are widely used [4], and in recent years, C-doped electrodes with excellent conductivity, such as graphene, have been introduced [5]. In contrast, the OERs are four-electron reactions, and the reaction mechanism is complex; many overvoltages are required in the O_2_-generating reaction [6]. Over the years, many researchers performed the experiments related to electrochemical oxygen evolution reactions (OER). Typically, noble metal oxides such as RuO_2_ [7] and IrO_2_ [8] have shown excellent catalytic activity and stability in both acidic and alkaline electrolytes. However, because of the scarcity and high cost of metals, there are limitations in usage for industrial applications. Therefore, considerable research efforts have been made to explore the inexpensive transition metal oxides toward OER catalysts [9,10]. Recently, research has been carried out on hybrid materials using transition metal dichalcogenide (TMD) based catalysts containing highly conductive carbon materials or nitrogen-doped multi-walled carbon nanotubes (NWCNTs) functionalized with nitrogen-rich emeraldine salts [11,12]. In addition, there have been active researches on electrocatalysts using colloidal synthesis to increase the electrical properties by controlling the composition, size, and conductivity of nanoparticles [13,14]. As a result, it is essential to develop an OER catalyst having low cost, high electrolysis efficiency, and excellent stability for an efficient energy conversion and storage system such as a fuel cell based on electrolysis of water. In particular, RuO_2_ and IrO_2_ precious metal oxides do not show long-term stability under alkaline electrolytic conditions [15]. Therefore, in order to produce a large amount of hydrogen at low cost, it is very important to develop a new catalyst material capable of lowering the OER overvoltage in the alkaline medium. Recently, cobalt-based composite materials [16], steel materials [17], and alloy materials have been studied as OER catalysts, and activity according to the modification of the catalyst surface and the pH of the medium has been studied [18]. The aim of this study is to develop a highly efficient OER catalyst with good stability based on inexpensive transition metals under alkaline electrolytic conditions. Here, two particles with different structures were combined to design an electrochemical OER catalyst for the efficient electrolysis of water. In recent years, heterojunction particles have exhibited various properties at the junction interface, such as an increase in the amount of surface active sites, electron-rich interface formation, effective charge separation, and suppression of exciton recombination; consequently, they have led to performance improvements in various application fields [19,20,21]. The following three points were considered when designing the OER electrode material: (1) the position of the bandgap of the two heterojunction bands was arranged to facilitate the flow of electrons; this can improve OER activity by lowering the activation energy of electron transfer to improve electrical conductivity [22]. (2) The voltage in the heterojunction particles was maintained above 1.2 V; in the application of the entire system of water electrolysis, the appropriate band gap of the material can balance the OER and HER. When voltage is applied from outside, electrons are transferred from VB to CB. The electrons in CB are transferred to the cathode, and the holes in VB are transferred to the anode, where the HER and OER reactions take place, respectively. However, if the band gap is too narrow, the cathode may be resisted and electrons may fall back to VB, reducing the reaction activity;. and (3) the active metals of the heterojunction particles were configured to have a redox reaction, taking into account the reduction potential. The relative difference in redox capacity of the active metals accelerates the reaction. The redox couple metals were investigated considering the reduction potential of the active metal, and the electron transport was facilitated by the heterojunction particles made of these redox couples. The redox couple metals selected in this study were Fe and Mn, which are the most inexpensive and abundant transition metals present in Earth’s crust. In addition, the mixed Fe/Mn oxide is an n-type semiconductor with a fermi level close to CB [23]. This causes easy electron transfer and can promote redox reaction. The metal oxides Fe_2_O_3_ and MnO were mixed, and the resulting material was used as an efficient, durable, and ecofriendly electrode catalyst for water electrolysis. During water electrolysis, it was expected that the redox reaction would proceed on its own—the ferrous metal with a slightly decreased potential value is reduced to Fe^3+^ → Fe^2+^, E_0_ = 0.77 V, while the Mn is oxidized to Mn^2+^ → Mn^3+^ (Mn^2+^ → Mn^0^, E_0_ = −1.18 V, too difficult)—and that the electronic cycle would be maintained continuously [24]. Pure Fe_2_O_3_ and MnO were synthesized via the sol–gel method, which is the simplest synthesis method, and an Fe_2_O_3_-MnO hetero-mixture grafted with the two oxides was also synthesized via the sol–gel method. Subsequently, the two types of metal-oxide powders were dispersed/coated on Ni foam (NF) to fabricate the final electrode. The objective of this study was to evaluate the synergy of Fe and Mn as a redox couple.

## 2. Experimental

### 2.1. Synthesis of MnO, Fe_2_O_3_, and Fe_2_O_3_-MnO Materials

MnO and Fe_2_O_3_ materials were prepared via the typical sol–gel method, as follows. For the synthesis of MnO, 4.0 g of manganese dichloride tetrahydrate (MnCl_2_·4H_2_O, ≥97%, Sigma-Aldrich., MO, USA) was dissolved in 100 mL of distilled (DI) water with stirring for approximately 2 h. The resulting homogeneous mixture was heated at 70 °C for 4 h to evaporate the DI water solvent and obtain a brown precipitate. The resulting precipitate was washed several times with water and ethanol alternately and dried overnight at 70 °C. MnO crystalline catalysts were finally synthesized via sintering at 500 °C for 4 h in an Ar gas atmosphere to prevent the Mn from being oxidized into trivalent or tetravalent ions.

Fe_2_O_3_ was prepared via a similar procedure. First, 6.0 g of ferric chloride hexahydrate (FeCl_3_·6H_2_O, ≥99%, Sigma-Aldrich., MO, USA) was dissolved in 100 mL of DI water. After approximately 2 h of stirring, ammonia water (NH_4_OH, ≥25% NH_3_ in H_2_O, Sigma-Aldrich., MO, USA) was added dropwise until the pH of the solution was 8.0. The orange precipitate was washed several times with DI water and ethanol and dried overnight at 70 °C. The resulting powder was sintered at 500 °C for 4 h to produce an Fe_2_O_3_ crystalline catalyst.

Finally, the heterojunction Fe_2_O_3_-MnO material was prepared via impregnation, as follows. The prepared crystalline Fe_2_O_3_ and MnO were mixed with absolute ethanol (CH_3_CH_2_OH, ≥99.8%, Sigma-Aldrich., MO, USA) at a weight ratio of 1:1, followed by stirring for 2 h to obtain a homogeneous solution. Then, a red-brown-colored powder was obtained through evaporation at 70 °C. The obtained powder was heat-treated at 400 °C for 3 h in an Ar atmosphere to produce the heterojunction Fe_2_O_3_-MnO crystal.

This study focused on identifying the synergistic effects of Fe and Mn. Although the manuscript did not include data, the OER reactions were already performed by varying the mass ratio of Fe_2_O_3_ to MnO (Fe_2_O_3_:MnO = 1:2, 1:1 and 2:1). As a result, the catalyst with 1:1 mass ratio exhibited the higher catalytic activity. This is probably due to the difference in particle size or shape of Fe_2_O_3_ and MnO, which may have disrupted the synergistic effect with other particles as more molecules aggregated among themselves. Therefore, when Fe_2_O_3_ and MnO were mixed in a 1:1 ratio, it was determined that the synergistic effect was increased with the most uniform and stable heterojunction. In other words, in order to clearly compare the performance of bimetallic and monometallic materials, only one Fe_2_O_3_-MnO material mixed in a 1:1 ratio was presented in this study and the synergy effect was clarified.

### 2.2. Fabrication of MnO/NF, Fe_2_O_3_/NF, and Fe_2_O_3_-MnO/NF Electrodes

The electrodes for OER were prepared as follows in Scheme 1a: The surface was treated with 5.0% HCl for approximately 30 min to remove surface impurities on the substrate, i.e., NF (1.0 cm × 1.0 cm). The surface of the NF was then washed with DI water and absolute ethanol and dried at room temperature. The coating solution was prepared by mixing 30 mg of the synthesized catalyst powder (MnO, Fe_2_O_3_, and Fe_2_O_3_-MnO) with 600 μL of ultrapure water, 600 μL of absolute ethanol, and 60 μL of a 5.0 wt.% Nafion solution (Sigma-Aldrich., MO, USA), followed by dispersion for 3 h. The coating solution was evenly coated onto the surface-treated NF using a dip-coating method and then dried at 70 °C. Subsequently, MnO/NF, Fe_2_O_3_/NF, and Fe_2_O_3_-MnO/NF electrodes were prepared via heat treatment at 200 °C for 3 h.

### 2.3. Characterizations

The crystal structures of the MnO, Fe_2_O_3_, and Fe_2_O_3_-MnO electrodes were examined via X-ray diffraction (XRD) analysis (Miniflex, Rigaku, Tokyo, Japan) using Ni-filtered CuKα radiation (30.0 kV, 15.0 mA) in the 2θ range of 20°–90°. The surface morphology of the particles was observed using a Hitachi S-4100 field-emission scanning electron microscope. Energy-dispersive X-ray spectroscopy (EDS) and EDS elemental mapping of the particles were performed using EDAX (EX-250, Horiba). Transmission electron microscopy (TEM) was performed using an H-7600 (Hitachi, Tokyo, Japan) instrument. X-ray photoelectron spectroscopy (XPS) was performed using an AXIS-NOVA (Kratos Inc., San Diego, USA) equipped with AlKα (1486.6 eV) radiation.

### 2.4. Electrochemical Measurements

All electrochemical measurements were performed at room temperature (25 °C) using an electrochemical cell test system and IVIUMnSTAT (Ivium Technologies, The Netherlands). The electrochemical OER activity was tested in an alkaline electrolyte (1 M KOH, pH = 14) using a standard electrochemical device (Scheme 1b): MnO/NF, Fe_2_O_3_/NF, and Fe_2_O_3_-MnO/NF were used as the working electrode, and Hg/Hg_2_Cl_2_ (saturated calomel electrode, SCE) and Pt wire were used as the reference and counter electrodes, respectively. The measured potential was converted into the potential of a reversible hydrogen electrode (V_RHE_) using the following calibration equation [25].

V_RHE_ = V_SCE_ + 0.241 + 0.059 × pH

For the initial stabilization of the electrode, 100 cycles of cyclic voltammetry (CV) scans (50.0 mV/s) were performed at approximately 1.0–1.7 V (vs. RHE). Linear sweep voltammetry (LSV) was performed at a scan rate of 5.0 mV/s in 1 M KOH. All electrochemical data was obtained with iR compensation by automatic current interrupt.

## 3. Result and Discussion

Figure 1A,B shows the XRD pattern and TEM image of the powder particles before the electrode assembly. The XRD pattern of MnO is consistent with the cubic structure with the Fm-3 space group, and the peaks corresponding to the (111), (200), and (220) diffraction planes are observed at 2θ = 34°, 40°, and 58°, respectively (JCPDS card no. 01-072-1533) [26]. The XRD peaks of Fe_2_O_3_ correspond to the diffraction planes attributed to the rhombohedral structure of the R-3c spatial group and were observed at 2θ = 24° (012), 33° (104), 35° (110), and 40° (JCPDS card no. 01-089-0598) [27]. The heterojunction Fe_2_O_3_-MnO particles exhibited the crystal-structure patterns of both MnO and Fe_2_O_3_. Unexpectedly, the intensity of the XRD peaks was increased overall for the heterojunction Fe_2_O_3_-MnO particles. This may be due to the further growth of crystals during the sintering process for grafting the two types of pure crystals. The peak corresponding to Fe_2_O_3_ was more distinct, suggesting that the Fe_2_O_3_ crystals entirely covered the MnO crystals. This is confirmed by the TEM image in Figure 1B; the MnO sample exhibited a broad and thin sheet shape (approximately 400–500 nm thick), and in the Fe_2_O_3_ sample, a thin cloth piece of approximately 100 nm was rounded to a 15 × 100 nm rod shape. Eventually, it appeared similar to a petal. In the heterojunction Fe_2_O_3_-MnO sample, a piece of Fe_2_O_3_ covered the MnO sheet, which is consistent with the XRD results.

Figure 2A shows the results of EDS analysis of selected surfaces of the film-type electrodes assembled using the manufactured powders. According to the EDS analysis results, the atomic ratio of Mn:O on the surface of MnO was approximately 1.0:1.2, and the atomic ratio of Fe:O on the Fe_2_O_3_ surface was approximately 2.0:3.3. This indicates that the metals were stoichiometrically contained in the particles and that the metal oxides were manufactured reliably. In the case of Fe_2_O_3_-MnO, the ratio of Mn:Fe:O was 1.2:1.0:5.5, although it was physically grafted. Figure 2B shows a scanning electron microscopy (SEM) image of the surface of the electrode coated with catalyst particles on the NF substrate. Regardless of the type of catalyst, the catalyst uniformly covered the NF. Compared with the powder catalyst form in Figure 1, the catalyst was evenly coated on the NF in its original shape, without deformation. Figure 2C shows the element mapping of the electrode surface, confirming that Mn, O_f_, and Fe were uniformly distributed. The intensity of the color is proportional to the concentration of the element, and the color intensity changes stoichiometrically. In particular, the color of Fe in the Fe_2_O_3_-MnO heterojunction is similar to that of Fe_2_O_3_, indicating that the Fe_2_O_3_ crystal covered the surface of MnO, which agrees with the results of XRD and TEM.

Figure 3 shows CV curves obtained when NF, MnO/NF, Fe_2_O_3_/NF, and Fe_2_O_3_-MnO/NF samples were used as the working electrode. Generally, NFs exhibit characteristic oxidation and reduction peaks of Ni at 1.4 and 1.3 V [28]. Repeated CV tests for 100 cycles were performed to evaluate the stability of the electrodes for oxidation–reduction. In addition, CV cycle repetition is necessary for the full activity of the electrode. As the number of CV scans increased, the oxidation and reduction peaks of the NF were shifted to a positive voltage, and after about 100 cycles the CV curve was constantly stabilized [29]. However, no potential change was observed in the MnO/NF, Fe_2_O_3_/NF, and Fe_2_O_3_-MnO/NF electrodes assembled here, despite cycling. This indicates that the MnO, Fe_2_O_3_, and Fe_2_O_3_-MnO catalysts formed a stable structure at the beginning, suggesting that they contributed to the activity of the catalyst. Notably, for the heterojunction Fe_2_O_3_-MnO catalyst, no peak change was observed during the 100 cycles, and the current density was more stable than those of the other two samples. This indicates that the MnO and the Fe_2_O_3_ particles were very stable and strongly bonded to each other; thus, there was little loss of catalytic active sites. The current density and peak area in the CV curve are closely related to the reversible redox reaction and the electrical active surface area [30]. In all NF-based catalysts, the peak area is expanded by increasing the volume of Ni(OH)_2_/NiOOH with cyclic repetition according to the following equations [31].

Ni^0^ + 2OH → Ni(OH)_2_ + 2e^−^

Ni(OH)_2_ ↔ NiOOH + e^−^ + H^+^

Furthermore, the overall current-density range for the NF electrode coated with the catalyst was significantly increased compared with that of pure NF, which may be related to the catalytic activity. A uniformly dispersed metal or metal oxide has been reported to provide a large number of active sites in the electrocatalytic reaction, resulting in a significant increase in the current density [32]. Both the current density and the peak area of the heterojunction Fe_2_O_3_-MnO/NF electrodes were significantly increased. This is presumably due to the excellent redox performance of Mn and Fe, and it is expected that the junction of the heterojunction structure generated more active sites and had a high electron density to promote electron transfer during the catalytic reaction [33].

The LSV curves were measured at the 100-cycle OER test. The results are shown in Figure 4A. In general, the activity of electrocatalysts is compared according to the overvoltage at a current density of 10 mA/cm^2^ in a water electrolysis reaction [34]. For a pure NF electrode, an overvoltage (η) of 430 mV was needed to reach a current density of 10 mA/cm^2^. In contrast, the MnO/NF and Fe_2_O_3_/NF electrodes exhibited relatively low over-voltages of 410 and 400 mV, respectively, indicating that they had better activity than the pure NF. The heterojunction Fe_2_O_3_-MnO/NF exhibited an overvoltage of 370 mV, confirming that it had the best OER activity among the electrodes examined in this study. Figure 4B shows the Tafel slopes derived from the LSV polarization curves. The Tafel slope indicates the inherent characteristics of electrocatalysts, and a smaller Tafel slope corresponds to better water oxidation reaction kinetics [35]. As shown in Figure 4B, the Tafel slope for the Fe_2_O_3_-MnO/NF electrode was 66 mV/dec, which was significantly smaller than those for the NF (100 mV/dec), MnO/NF (74 mV/dec), and Fe_2_O_3_/NF (78 mV/dec) electrodes. The stability of the electrocatalyst is very important for the long life of the water electrolysis system. In order to assess the durability of the MnO/NF, Fe_2_O_3_/NF, and Fe_2_O_3_-MnO/NF electrodes, a chronopotentiometry test was performed for 20 h, and the applied current density was set to 10 mA cm^−2^. Figure 4C shows the chronopotentiometric curves of water electrolysis for different catalyst electrodes with a constant current density of 10 mA cm^−2^. The potentials required to achieve a current density of 10 mA cm^−2^ at the MnO/NF, Fe_2_O_3_/NF, and Fe_2_O_3_-MnO/NF electrodes were ~1.64, 1.63, and 1.60 V, respectively; the potential was maintained the same throughout the 20 h electrolysis reaction. Looking closely at the results of the first two hours of the reaction of all the electrodes, the catalyst was stably activated as the potential gradually decreased. This showed that the NF used as the substrate electrode was partially exposed to improve the electrical conductivity of the catalyst and can easily transfer the current density. Jaramillo et al. [36] reported that OER stability losses could be due to many factors such as corrosion, material degradation, surface passivation, and so on. As demonstrated in the SEM image of the electrode surface after OER, the catalyst developed in this study was stably bound to the NF surface without any leaching or collapsing during the reaction. Thus, these results show that Fe_2_O_3_-MnO/NF electrodes can exhibit good electrochemical durability and excellent OER activity in practical alkaline aqueous solutions.

Figure 5A,B shows the LSV curves at 1000 cycle for the OER and SEM images of the electrode surface before and after the 1000 cycled OER when the heterojunction Fe_2_O_3_-MnO/NF was used as the working electrode. Surprisingly, when the Fe_2_O_3_-MnO/NF electrodes were used, the LSV curve retained its performance as early as in the 1000-cycle OER test. The Tafel plots obtained after 1000 cycles exhibited negligible slope differences, indicating that the Fe_2_O_3_-MnO/NF electrode had very good durability. Figure 5B shows an SEM image of the Fe_2_O_3_-MnO/NF electrode surface before and after 1000 cycles in the OER. On the surface of the NF electrode after the OER, the Fe_2_O_3_-MnO heterojunction particles were well attached equally before the reaction, and they surprisingly had a similar size and shape to the fresh Fe_2_O_3_-MnO particles. This is presumably because the heterojunction structure was well maintained owing to the self-redox reaction of Mn and Fe, which facilitated the electron transfer, and the voltage application did not damage the NF electrode as a substrate.

Figure 6 shows the XPS results of Fe2p, Mn2p, Ni2p, and O1s for the Fe_2_O_3_-MnO/NF electrode after the OER. An Ni2p_3/2_ peak corresponding to metal Ni appeared at 852.6 eV, and an Ni2p_3/2_ peak corresponding to Ni^2+^ appeared at 855–856 eV [37]. Unfortunately, no complete metal-Ni component was observed in the NF electrode before the OER. This is evidence that the NF was partially oxidized during the process of coating the particles on the NF. Perhaps the OER performance could have been improved if the NF used as the substrate electrode had maintained a perfect metal state. However, only the Ni2p_3/2_ highest occupied molecular orbital peak attributed to Ni^2+^ was broadly observed at 856.1 eV for NF before the reaction (fortunately, NF was not completely oxidized to NiO, as lowest unoccupied molecular orbital (LUMO) moiety was not observed). After the OER, the binding energy shifted slightly to a higher value (856.4 eV), and the peak shape changed sharply. Furthermore, the Ni2p_1/2_ peak corresponding to the LUMO was observed at 874.1 eV after the reaction. This indicates that some of the Ni in the NF was oxidized to Ni^2+^ during the OER. The Fe2p_3/2_ peak for fresh Fe_2_O_3_-MnO/NF was separated into peaks in various oxidation states, and peaks attributed to Fe^0^, Fe^2+^, and Fe^3+^ were observed at binding energies of 706.4, 708.5, and 711.2 eV, respectively [38]. Additionally, the Fe2p_1/2_ peak corresponding to Fe^3+^ was observed at 724.9 eV. The observation of Fe^0^ and Fe^2+^ in the XPS analysis indicates that the Fe^3+^ accepted electrons from the NF, becoming reduced and oxidizing the NF, during the loading of the Fe_2_O_3_-MnO particles onto the NF. However, after the OER, the isolated peaks converged to one peak at 711.2 eV. This suggests that the oxidation state of Fe during the OER was stable at 3+ [39]. In the high-resolution Mn2p XPS spectrum obtained before the OER, two main spin-orbitals corresponding to Mn2p_3/2_ and Mn2p_1/2_ (indicating the presence of Mn^2+^) were observed at 641.6 and 653.6 eV, respectively [40]. Additionally, the Mn2p_3/2_ peak attributed to Mn^3+^ exhibited a low binding energy (644.7 eV). After the OER, the Mn2p_3/2_ peak corresponding to Mn^2+^ converged at 641.6 eV as a single peak, the intensity of the peak decreased, and the peak width increased, similar to the results for Fe2p. This indicates that the oxidation state of Mn was stabilized to 2+ during the OER. According to the XPS results for Fe2p and Mn2p, the active metals that constituted the electrode during the OER were reduced to Fe^3+^ → Fe^2+^ (Mn^2+^ → Mn^3+^) when charged, and the electrons moved in the opposite direction during discharging. At this time, redox couples were formed between Fe^3+^ and Mn^2+^, and they were repeatedly oxidized–reduced and then restored. It was confirmed that their redox reaction was fairly stable and that the OER performance continued to be maintained after 1000 cycles, indicating a synergy caused by the coexistence of Mn and Fe. Before the OER, the XPS spectra of O1s were separated by various binding energies at 529.9, 531.9, and 533.8 eV, which can be attributed to M–O, H_2_Oads, and M–OH, respectively [41]. After the OER, the peaks were slightly shifted to higher binding energies. The H_2_Oads peak because of adsorbed water was stronger than the M–O peak corresponding to O_2_ in the lattice. This is because during the OER, when the active metals underwent repeated oxidation–reduction reactions, a part of the M–O bond of the lattice was broken. The ratio of H_2_O_ads_ peak and lattice M–O peak after OER changed significantly. It can be expected as a result derived from the partial participation in the reaction in which the surface lattice oxygen atoms of the metal oxide are exchanged with the oxygen atoms of the electrolyte solution. OH^-^ species adsorb to MO via one electron transfer, and then O–O bonds of MO(OH) are formed. Here, additionally, OH^−^ species were adsorbed to generate MOO(OH). At this time, a portion of the surface lattice oxygen may be released as the higher valent oxide is transferred to the lower valent oxide. It is believed that the OER and the redox transition of the transition metal oxide in the alkaline medium are correlated. Moreover, recent research papers [42] have demonstrated that oxidation of lattice oxygen in metal oxides can enhance the OER activity and induce non-concerted proton–electron transfer.

However, this did not appear to be a major factor affecting the oxidation state of Mn or Fe; rather, it may have been related to the partially damaged NF electrode.

## 4. Conclusions

Fe_2_O_3_ and MnO were synthesized via a sol–gel method, and an Fe_2_O_3_-MnO heterojunction mixture was prepared. These particles were coated onto NF electrodes, and the electrochemical properties of the resulting materials were evaluated. XRD patterns and TEM images of the powder particles were analyzed to confirm their crystallinity and shape. Through SEM and elemental mapping of the NF surface after a 1000-cycle OER test, the adhesion degree of the particles on the NF was examined. The change in and stability of the oxidation state of the active metals were determined based on the XPS spectra before and after the OER. As a result of the OER, a lower overvoltage (1.60 V/10 mAcm^−2^) was obtained for the Fe_2_O_3_-MnO/NF (containing the heterojunction of the two oxides) than for the pure Fe_2_O_3_/NF and MnO/NF, and the Tafel slope for the heterojunction electrode was as small as 66 mV/dec. Even after 1000 cycles, the OER performance of Fe_2_O_3_-MnO/NF remained stable, without deterioration. These results indicate that during Mn and Fe self-redox reactions in Fe_2_O_3_-MnO/NF hetero-electrodes, the Mn oxidizes O_2_ in water and the Fe reduces H_2_, facilitating the migration of electrons. Eventually, it appears that this is a synergistic effect in the OER.

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
