# Peer review of "Electrochemical Synergies of Heterostructured Fe2O3-MnO Catalyst for Oxygen Evolution Reaction in Alkaline Water Splitting"

_nanomaterials, 2019, doi:10.3390/nano9101486_

Round 1

Reviewer 1 Report

In this article, Kim et al. presented a new nanomaterial based on Fe2O3-MnO for oxygen evolution reaction. The evaluation was done by means of typical characterization techniques, obtaining long-term stable features. I agree that this new approach is interesting for researcher investigating fuel cell. However, the novelty of this work is not clear since the results obtained have not been compared with the state of the art. Other technical issues:

1) Better description of the state of the art should be included. Here, some examples:

ACS Appl. Mater. Interfaces2018103731300-31311 and ACS Appl. Mater. Interfaces201684329461-29469.

2) Explanation of the weight ratio choses (i.e. 1: 1)

3) Catalyst loading  and normalize all CVs to the loading

4) Better justification of the heterojuntion between Fe2O3 and MnO

5) Real application of the material in fuel cell

6) Include in the scheme 1b the reference electrode used.

Author Response

Thank you very much for your comment. Your comment was very helpful in improving the manuscript. "Please check the attached file for answers and corrected manuscripts." We mark the additions and modifications to the manuscript in red. We sincerely hope that our manuscript can be published in the Nanomaterial journal.

Reviewer 2 Report

This in an interesting manuscript with both synthesis and characterisation of hetero-junction band of Fe and Mn oxides.

Data are reported with accuracy and conclusions are supporter by the results obtained by the authors by jeans of different experimental techniques.

In my opinion the manuscript deserves ti be' published as is in Nanomaterials.

Author Response

Many thanks to Reviewer for accepting our manuscript for publication

Reviewer 3 Report

The manuscript presents a systematic study on the use of metal oxide materials to promote oxygen evolution reaction. It is observed certain shift of the onset potential of OER due to the formation of a semiconductor heterojunction. The paper is easy to read and focuses mostly on material characterization. Few comments are indicated below:

L45-49: Provide a brief sentence that justifies the three selection criteria for the design of OER. Why Fig. 2 does not show the absence of Fe in MnO/NF and the absence of Mn in Fe2O3/NF? L157-160- Which stable phase? Please develop further. L169-163- Do these results also demonstrate that the coating completely covers the NF substrate avoiding interfacial reaction between the Ni substrate and the solution as observed in the blank (uncoated foam)? There is a notorious reduction on the OER potential when using the heterojunction electrode. However, it is also observed the oxidation peak close to 1.4 V similar to the one observed to the NF, but not observed for the other coated electrodes. Is there any explanation that may be given? Why the LSV do not show the peaks observed in the CV analysis for the same materials? This is odd. L247-249- How is the Me-O bond broken? What is the energy/cost impact that may be expected if this material is used? Long performance of Pechini or sol-gel coated electrodes is usually compromised due to the weak attachment of the coating to the substrate. This results in the coating wearing off during gas evolution. Even though the authors state that the structure of the materials is the same, have had the authors observed any leaching to solution during operation? The manuscript would benefit from quantitative data on the comparative oxygen production using different electrode materials.

Author Response

(The authors gave the same response as above.)

Reviewer 4 Report

Comments to “Electrochemical synergies of heterostructured Fe2O3-1 MnO redox couple for oxygen evolution reaction in 2 alkaline water splitting”

Description:  Fe2O3, MnO and heterojunction Fe2O3-MnO particles have been synthesized and (firmly attached to Ni-foam) exploited as an OER electrode in alkaline medium. When compared Fe2O3-NF, MnO-NF with Fe2O3-MnO-NF systems the authors found out that the heterojunction structured material shows the best OER properties. The work shows some interesting aspects and should be considered for publication after revision.

Further lab work is required; revision required- subject to  revision.

Regarding the title: Fe2O3-1 MnO redox couple; A redox couple consist of the oxidized and the reduced form of one and the same (e.g. catalytic active) species. Thus, to my understanding Fe2O3 and MnO cannot form a redox couple in the direct sense of the words. This represents a scientific incorrectness which is not acceptable for any scientific journal. Page 2 line 47-49 “…the active metals of the heterojunction particles were configured to have a redox reaction, taking into account the reduction potential….” It is not clear to me when the elements are supposed to have a redox reaction. During mixing of the oxides at low temperatures, or when the powder was heated to 400 °C thus when they form the heterojunction or during catalysis of the OER? Please specify this more clearly. Has voltage drop compensation been applied to the electrochemistry data? The authors should add OER measurements carried out under steady state conditions, as e.g. chronopotentiometry testing with duration at least 20 hours. Introduction part: Steel has gained a come-back as a highly active and stable OER electrocatalysts e.g. in alkaline media (see e.g. Schäfer et al., Energy Environ. Sci., 2015, 8, 2685-2697; Schäfer et al., Energy Environ. Sci., 2016, 9, 2609-2622) as was recently reviewed in an article (ACS Energy Lett.2018, 33, 574-591). The authors may take into consideration to cite these articles. Abbreviations should be explained where they appear first: abstract line 3: Ni-foam (NF)

Author Response

(The authors gave the same response as above.)

Round 2

Reviewer 1 Report

The manuscript has been improved substantially. I recommend this article for publication in the present form.

Author Response

We greatly appreciate your kind comments. Thank you.

Reviewer 3 Report

Authors conducted a thorough revision. Acceptance is suggested.

Author Response

(The authors gave the same response as above.)
